# On the surface or down below: Field observations reveal a high degree of surface activity in a burrowing crayfish, the Little Brown Mudbug (*Lacunicambarus thomai*)

**Kaine M. Diehl**[ID]**, Nicoleena M. Storer**[ID]**, Hogan D. Wells, Destinee A. Davis**[ID]**, Zachary J. Loughman, Zackary A. Graham**[ID] *

Department of Organismal Biology, Ecology & Zoo Science, West Liberty University, West Liberty, WV, United States of America

* Zackary.graham@westliberty.edu

## Abstract

Opposed to most crayfish species that inhabit permanent bodies of water, a unique burrowing lifestyle has evolved several times throughout the crayfish phylogeny. Burrowing crayfish are considered to be semi-terrestrial, as they burrow to the groundwater—creating complex burrows that occasionally reach 3 m in depth. Because burrowing crayfishes spend most of their lives within their burrow, we lack a basic understanding of the behavior and natural history of these species. However, recent work suggests that burrowing crayfishes may exhibit a higher level of surface activity than previously thought. In the current study, we conducted a behavioral study of the Little Brown Mudbug, *Lacunicambarus thomai* using video surveillance to determine their degree of surface activity and behavioral patterns. Throughout 664 hrs of footage, we observed a surprisingly high amount of activity at the surface of their burrows—both during the day and night. The percentage of time that individual crayfish was observed at the surface ranged from 21% to 69% per individual, with an average of 42.48% of the time spent at the surface across all crayfish. Additionally, we created an ethogram based on six observed behaviors and found that each behavior had a strong circadian effect. For example, we only observed a single observation of foraging on vegetation during the day, whereas 270 observations of this behavior were documented at night. Overall, our results suggest that burrowing crayfishes may exhibit higher levels of surface activity than previously thought. To increase our understanding of burrowing crayfish behaviors ecology, we encourage the continued use of video-recorded observations in the field and the laboratory.

## Introduction

Crayfish are a diverse group of freshwater crustaceans with over 700 species globally [1, 2]. Many crayfishes are considered to be model organisms for behavioral biology due to their ease

**Data Availability Statement:** The data are available at Dryad at the following DOI: https://doi.org/10.5061/dryad.kh189328r.

**Funding:** The author(s) received no specific funding for this work.

**Competing interests:** The authors have declared that no competing interests exist.

of capture, suitability for laboratory studies, simple neural architecture, and complex behavioral repertoires [3–7]. However, this work stems from studies of a few model species from North America (i.e., *Procambarus clarkii*, *Procambarus virginalis*, *Faxonius virilis*); all of which inhabit lentic and lotic aquatic environments. Although most crayfish species inhabit surface water systems like streams, lakes, rivers, ponds, and marshes, crayfishes have repeatedly evolved a semi-terrestrial burrowing lifestyle throughout their phylogeny [8–10]. Despite all species of crayfish possessing the ability to burrow to some degree [11], the complexity and reliance on these burrows vary greatly both within and between species [12, 13].

Crayfishes have historically been assigned to three different ecological classifications based on their burrowing ability and their reliance on their burrows: tertiary, secondary-, and primary burrowing species [12, 14]. Tertiary burrowing crayfishes are species which rely on permanent bodies of water and are only capable on constructing relatively simple burrows underneath benthic substrates. Secondary burrowing species can also inhabit permanent bodies of water, but during periods of drawdown they are capable of excavating complex burrows to access groundwater. Primary burrowing species (hereafter referred to as burrowing crayfish) are often considered semi-terrestrial as they are highly reliant on their burrows for their entire life; as they potentially serve as a location for breeding, refuge from predators, and as a location to acquire and cache food [12, 13]. Burrowing crayfish burrows are often extremely complex and may have several entrances (i.e., portals), tunnels, and chambers that can reach several meters deep [12, 13]. In addition to providing shelter for the crayfish that construct them, such burrows can serve as necessary habitat for other organisms [15–18] and increase soil bioturbation and habitat complexity [19–21].

Compared to our knowledge of tertiary and secondary crayfishes that occur in lentic and lotic environments, our understanding of burrowing crayfish behavior is scant [13, 22]. Historically, burrowing crayfish were thought to spend nearly their entire lives below the surface, and thought to occasionally leave their burrows for brief terrestrial excursions to search for food or to find a mate [12]. Minimal surface activity and their propensity to retreat to their burrow when startled has led to difficulties in capturing and observing burrowing species [23, 24]. Resultant of their reclusive lifestyle, our understanding of burrowing crayfish activity patterns and behavior is minimal, with much of the work on these species being derived from *in situ* natural history observations [12, 25–29] and not systematic investigations. However, despite these difficulties, semi-naturalistic mesocosms and custom-built observation chambers have been useful in the study of burrowing crayfish behavior [30–34].

Burrowing crayfish are considered to be nocturnal [35], which further complicates our ability to adequately study their behavior. But in some instances, burrowing crayfish are observed at the surface of their burrow during the daytime [12, 35–37]. Interestingly, recent studies reveal that burrowing crayfish may exhibit a higher level of surface activity than previously thought [36, 38]. Bearden et al. (2020) conducted the most comprehensive study to date on the activity of burrowing crayfish [38]. Using motion-trigger laser photography in two species of burrowing crayfish (*Lacunicambarus erythrodactylus* and *Procambarus holifieldi*), Bearden and colleagues measured activity throughout a one-year period. In their study, activity was considered based on whether the crayfish was visible at the entrance of their burrow [38]. This study confirmed that nocturnal activity occurred most frequently, specifically during times of cool groundwater temperature and warm air temperatures [38]. Furthermore, despite observing substantial activity at night for both species, there were many occurrences of daytime surface activity throughout their study.

Bearden et al. (2020) provides robust evidence regarding the environmental correlates of activity in these two species and serves as a solid foundation to build off for future studies on burrowing crayfish surface activity. However, the use of photography and not videography

limited their insights into crayfish surface behaviors. As such, the photographs made defining behaviors difficult, and limited behavior determination to chimney construction and surface activity [38]. As stated previously, Bearden et al. (2020) goals were not specific to defining burrowing crayfish surface behaviors, but rather determining if surface activity was correlated to specific environmental correlates. Therefore, the behavioral complexity of burrowing crayfishes still remains unknown. Given their utilization of complex burrow systems and evidence of social behavior [13], burrowing crayfishes species may exhibit complex ritualistic behaviors like non-burrowing crayfishes [3, 13, 39]. However, no study to date has conducted long-term behavioral observations of a burrowing species in a completely natural environment.

Here, we conducted a behavioral study of a burrowing crayfish species through video-recorded field observations. Specifically, we monitored the surface of burrows inhabited by the Little Brown Mudbug, *Lacunicambarus thomai* (Fig 1). *Lacunicambarus thomai* is as a burrowing crayfish species with a high propensity to inhabit burrows in marshes, roadside ditches, and flooded fields [40]. Populations of *L. thomai* often live in localized colonies with conspecifics and inhabit burrows that are relatively simple but can nonetheless be up to 1–1.5 m deep [40, 41]. Compared to most burrowing crayfish species, *L. thomai* and other members of the genus *Lacunicambarus* seem to exhibit a high degree of surface activity [36, 37, 40–44], which makes them ideal for our investigations. Overall, we hope to shed light on the surface behavior of *L. thomai* by 1) determining the prevalence of surface activity and 2) creating an ethogram specific to burrowing crayfish, and 3) analyzing the behavioral repertoire and prevalence of behaviors exhibited by *L. thomai*.

## Methods

### Study site

In June and July 2020, we conducted non-stop 24-hour video surveys within a population of the Little Brown Mudbug, *Lacunicambarus thomai* (Fig 1) in Hickory, Pennsylvania, U.S.A. This population of *L. thomai* is located on the edge of a man-made pond on a private, residential yard. More specifically, our study site was located in a dense patch of grasses on the edge of the pond with no canopy cover. Within this location, the water table remains shallow throughout the year and there is an estimated population of 20 *L. thomai* within an area of ~20 m$^2$. Burrows were sporadically located throughout the population and were ~1 m to ~ 5 m from one another. No other crayfish species have been collected or identified from this location and no crayfish have ever been collected from the pond. We chose which burrows to record based on evidence of recent surface activity (activity observed at burrow portal during preliminary night-time observation), as well as whether the burrow portal was visible from an overhead view. Because capturing burrowing crayfish often destroys the burrow and the population was relatively small, we chose to not capture the individuals. Therefore, we do not report morphological or demographic data from the recorded footage (i.e., sex, body size). However, based on the circumference of the burrow tunnels as well as the video footage, all burrows in which we recorded (see below) were from adult *L. thomai*. Importantly, although other burrowing crayfish species are known to exhibit social behavior and have multiple individuals within a single burrow, *Lacunicambarus* spp. seldomly exhibit social behavior and therefore it is assumed that each burrow only housed a single crayfish [41, but see 42].

Because our work was conducted on privately owned property on an unlisted crayfish species and we did not collect any specimens for this project, no specific permits were necessary for our research.

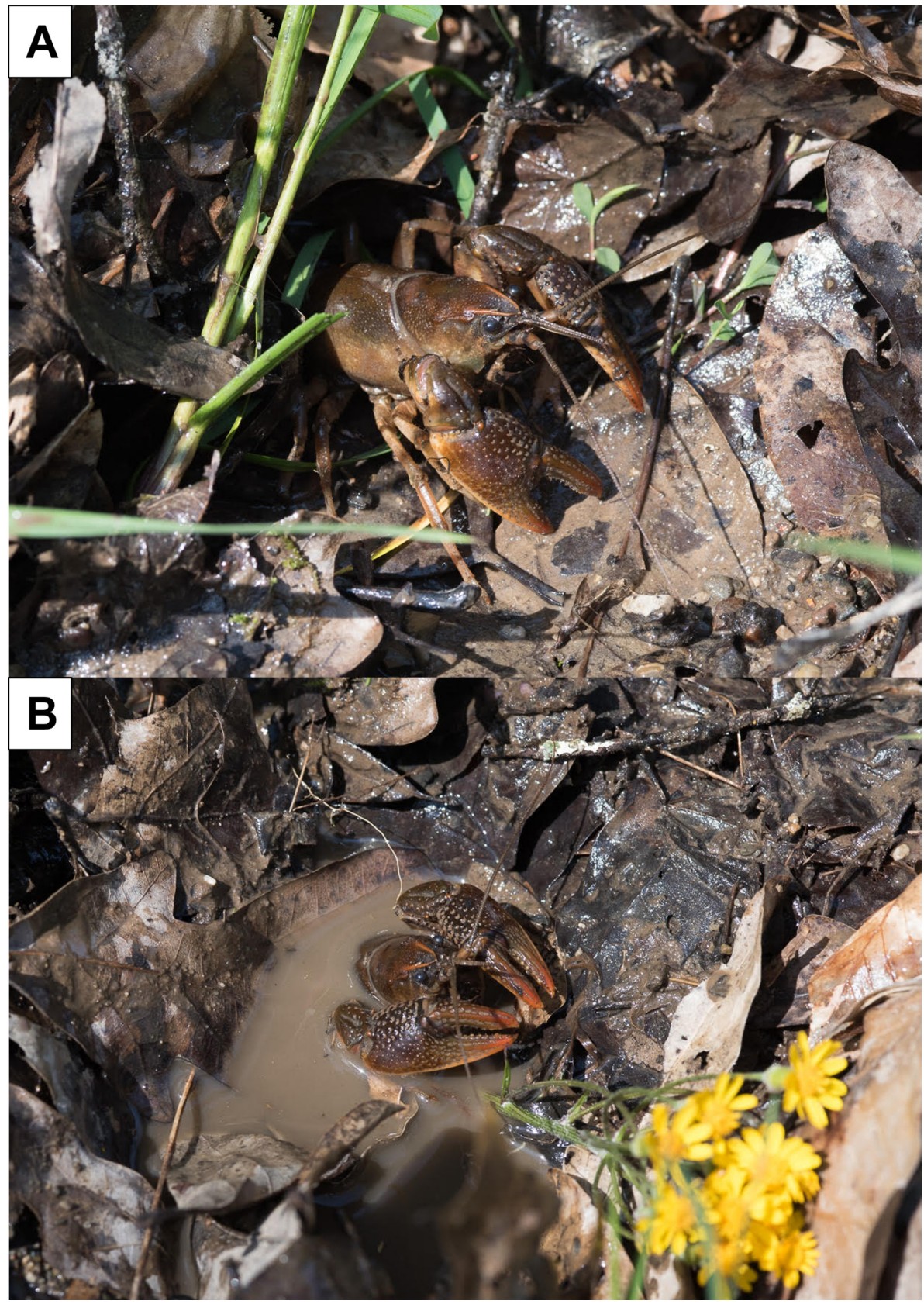

**Fig 1. Photographs of adult burrowing crayfish, the Little Brown Mudbug,** *Lacunicambarus thomai*, **exhibiting daylight activity outside of its burrow.** The burrow's entrance portal is out of sight due to the extensive leaf litter coverage. Photographs courtesy of John Freudenstein.

## Videography

Overtop each burrow that we recorded, a custom-built video camera mount was constructed by placing a Reolink 5MP video camera inside of a 5-gallon bucket that we mounted with wire approximately 0.5 m above the burrow (Fig 2). These camera mounts protect the camera from the external environment and reduce glare. We recorded each crayfish until their burrow was either capped or until there was a video camera malfunction. If a crayfish burrow contained more than a one portal, we only mounted the video camera overtop the larger, primary portal which is where burrowing crayfish exhibit nighttime activity (*Z. Loughman*, personal communication). In total, we recorded and reviewed (see below) 664.7 hr of footage from six adult *L. thomai* (Table 1). The length of time recorded for each crayfish ranged from 44.5 6 hr to 263.82 hr (Table 1).

## Ethogram

We reviewed all video footage with Windows Movie Editor Software. While reviewing the footage for each crayfish, we created an ethogram based on observed behaviors (S1–S3 Files). In total, our ethogram contained 5 surface activity behaviors, as well as a single "inactive" behavior (under) which was used to denote when the crayfish was not visible at the surface of the burrow. See Table 2 for detailed descriptions of each behavior. Then, using our ethogram, we recorded what behavior each crayfish was exhibiting, the duration of the behavior, and the time in which the crayfish started and ended the behavior. We also noted whether each crayfish exhibited behaviors during the day or night based on the civil twilight time listed for Hickory, Pennsylvania, U.S.A (collected from www.time.unitarium.com). For our study, we considered a crayfish to be active if they were visible at the surface of their burrow (Fig 3; [35]). We considered a crayfish to be considered inactive if they were not visible at the surface of their burrow. We are unable to report on any below-surface activity within our study.

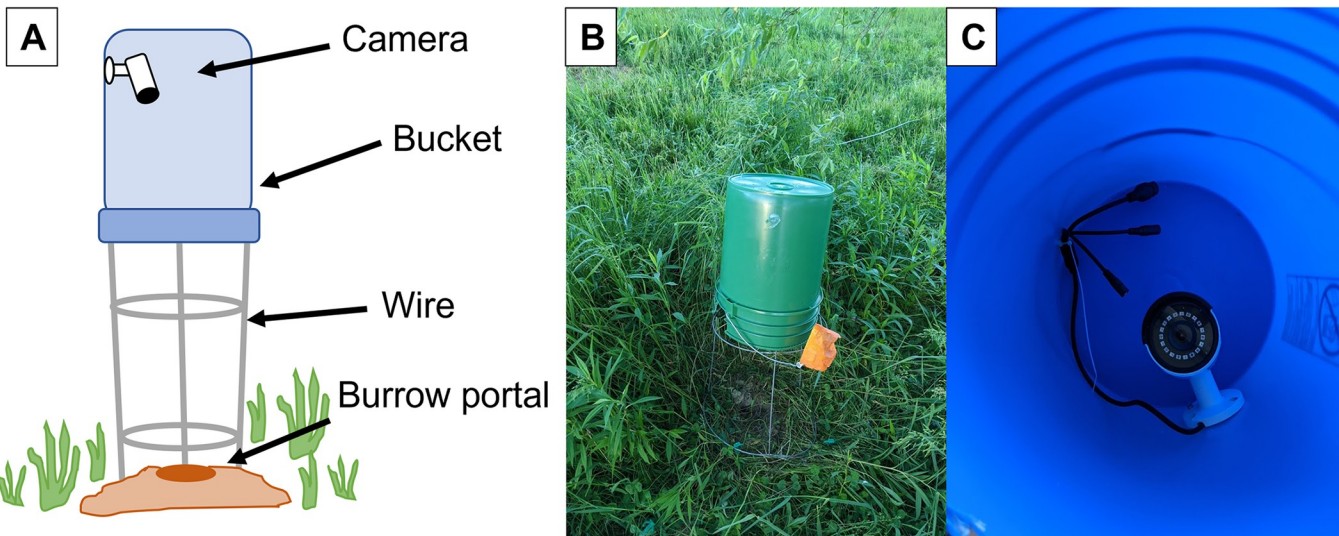

**Fig 2. Images representing our custom video surveillance mount.** (A) Diagram of our video surveillance setup used to monitor crayfish behavior. (B) Photograph of the setup *in situ* **C**) Photograph of the view of the camera mounted from inside of the bucket.

**Table 1. Dates and total time that each crayfish was filmed throughout the duration of our study.** Only a single crayfish (Crayfish 5) was filmed discontinuously due to failure of the video recorder.

| Crayfish | | |
|---|---|---|
| | Dates Filmed (MM/DD/YYYY) | Total Time Filmed (hr) |
| 1 | 06/03/2020–07/02/2020 | 44.56 |
| 2 | 07/02/2020–7/20/2020 | 263.82 |
| 3 | 06/28/2020–06/30/2020 | 40.34 |
| 4 | 07/01/2020–07/06/2020 | 105.93 |
| 5 | 06/13/2020–06/14/2020; 06/28/2020–07/03/20 | 151.73 |
| 6 | 07/21/2020–07/24/2020 | 57.35 |
| Combined | - | 664.73 |

## Environmental data

Because we wanted to investigate broad-scale trends of how the surface activity of *L. thomai* varied with environmental conditions, we gathered environmental data from an online source (www.weatherunderground.com). We collected data from the closest available weather station in Imperial, Pennsylvania, U.S.A (~24 km away). Specifically, the environmental data that we collected from this source was air pressure, precipitation, temperature, and humidity. Later, these variables were used as covariates to determine which environmental variable best predicted *L. thomai* surface activity.

## Statistical analysis

We conducted all statistical analyses in R version 3.5.1 [45]. For all of the models described below, we visually inspected our data using the performance package in R to assess normality and collinearity of our model fits [46]. For descriptive analyses of *L. thomai* behavior, we calculated the frequency, mean, and range of each behavior. Additionally, we wanted to describe the activity of *L. thomai* based on the time of day. Because of the uneven distribution of video footage length for each crayfish (Table 1), we decided to report the majority of our results in terms of percentages, and not raw values. Therefore, we calculated the percentage of crayfish which exhibited surface activity at each hour by dividing the number of hourly observations in which surface activity was observed by the total number of hours which we recorded throughout our entire study. We averaged these values together for all crayfish which enabled us to investigate patterns in surface activity based on the hour of the day. The same procedure was conducted to determine the percentage of time that each crayfish was observed at the surface throughout the daytime and nighttime.

**Table 2. Ethogram used to quantify the behaviors from recordings of *Lacunicambarus thomai*.**

| Behavior | Description |
|---|---|
| Rest–Claws Open | Crayfish is motionless and partially out of burrow portal with both claws in a "U" orientation directed anteriorly. Movements consist of simple shifts in body position. |
| Rest = Claws Joined | Crayfish is motionless and resting on burrow portal lip with both claws in a "V" orientation. Both claw tips touching or nearly touching directly anterior to the rostrum. |
| Forage | Crayfish is actively manipulating and foraging for plant material (leaves, grass, or roots). |
| Excavate | Crayfish is actively carrying, moving, or placing burrow substrate outside of the burrow beyond the burrow portal. |
| Hunt | Crayfish quickly rushes beyond the burrow portal in attempt to predate on living animal prey. |
| Under | The crayfish is inactive on the surface and not visible. |

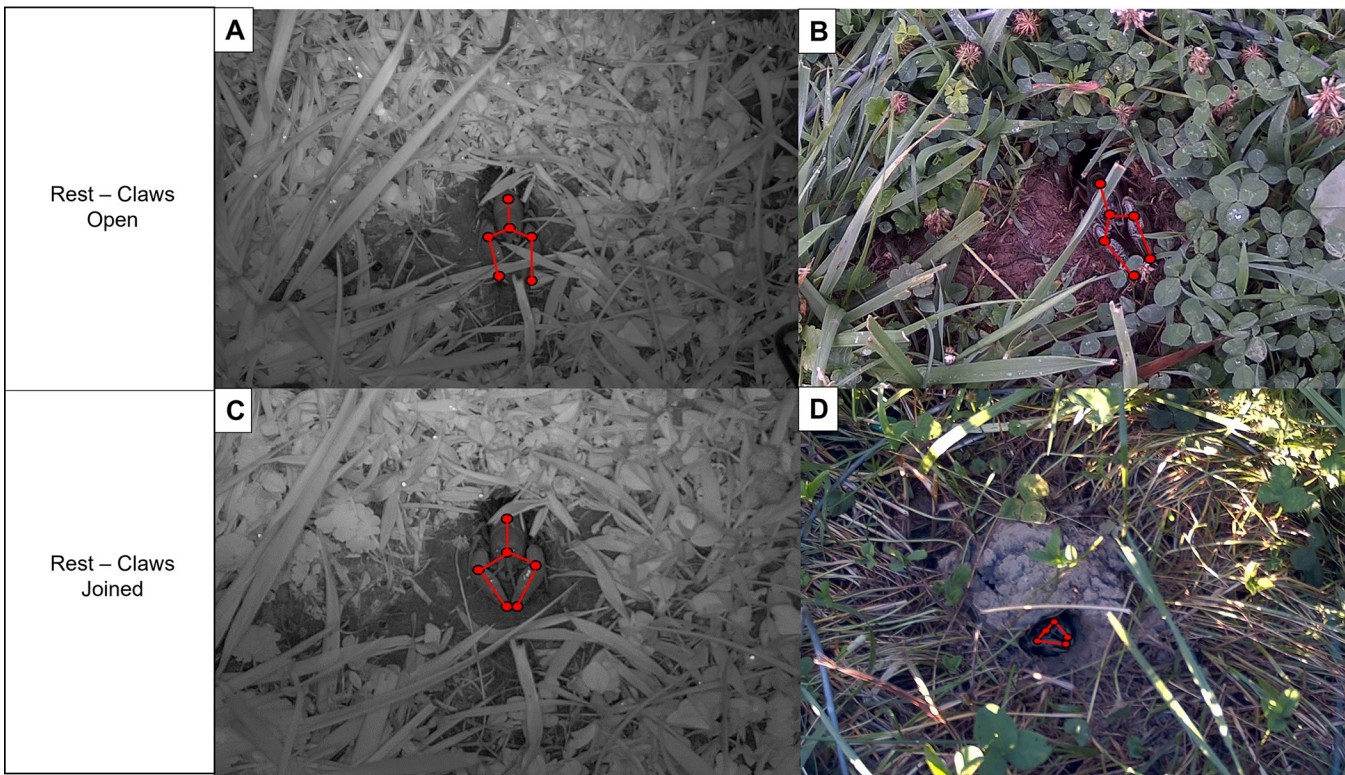

**Fig 3. Example of rest–claws open and rest–claws joined behaviors from video surveys *of Lacunicambarus thomai*.** The top two images (A) and (B) represent crayfish in the rest–claws open posture, whereas the bottom two image (C) and (D) represent the crayfish in the rest–claws joined posture. Red circles and red lines are presented to give a representation of the differences in articulation between the claws open and the claws joined postures. Photos (A) and (C) were taken at night, whereas photos (B) and (D) were taken during the day.

To determine which environmental variables best related to burrowing crayfish activity, we initially used a model selection procedure with several linear mixed effect models with activity as dependent variable and the environmental variable (i.e., air pressure, precipitation, temperature, and humidity) as the independent variable. Visualizations of the environmental variables are depicted in the Supplemental Information (S1–S4 Figs). For this analysis, we determined a crayfish to be active during a given hour if they were recorded at the surface of their burrow within any point during that hour. In these models, we included crayfish ID as a random effect. Based on the Akaike information criterion (AICc) of these models [47], we found that only temperature and humidity effected activity of burrowing crayfish, and therefore we proceeded with these two variables for the remainder of this analysis.

Thus, to evaluate how environmental variables (i.e., humidity and temperature) and time of day (i.e., hour) influenced activity of *L. thomai*, we used multimodal averaging to determine how temperature, humidity, and time (independent variables) influenced the activity of crayfish (dependent variable). For this analysis, we determined a crayfish to be active during a given hour if they were recorded at the burrow portal within any point during the hour. Additionally, because there were multiple observations from each crayfish, we included crayfish ID as a random effect to control for individual differences in individual activity.

We first fit the full model to the data with the *lme4* library of the R statistical package [48]. The full model contained temperature, humidity, time, and the interaction between these three variables as the independent variable. Because our dependent variable (i.e., activity) was coded as 1 (active) or 0 (inactive) we used a binomial distribution of error in this model. This

model also included crayfish ID as a random variable. Then, we fit models that contained a subset of the terms in the full model. For each model, we calculated the corrected Akaike information criterion and the Akaike weight (i.e., the probability that the model described the data better than the other models in the set). Finally, we used the parameters of each model and Akaike weight to calculate a weighted average of each parameter. These model-averaged parameters enable one to determine the expected probability for any combination of values for the fixed factors in the full model. This approach eliminates the use of *P* values because all models (including the null model) contributed to the expected probabilities. Importantly, collinearity was unlikely to bias our model averaged parameters [49]; because none of our independent variables (humidity, temperature, and time) were not strongly correlated (all $r^2 <$ 0.6).

To determine whether each specific behavior was more or less likely to occur during the day or night, we conducted six separate Chi-square tests of independence. We conducted 5 separate chi-squared tests for each individual behavior (relax, guard, forage, excavate, hunt, and under). We also performed a single chi-squared test for all six behaviors combined to determine whether all activity was more likely to occur during the day or the night. Additionally, we analyzed the likelihood of each behavior occurring during the day or night with a contingency mosaic plot [50]. Continency mosaic plots display the Pearson residual values as colors in relation to the deviation between expected (equally as likely to occur during the day and night) versus the observed frequencies [50]. This allows us to both visually and statistically investigate the directionality of the observed frequencies of each behavior (and the combined behaviors) in relation to equal frequencies.

## Results

### Hourly surface activity

We observed *L. thomai* exhibiting surface activity throughout every single hour of the day. However, there was variation in the times that *L. thomai* were most likely to be observed active at burrow portals. Surface activity peaked from 20:00 hr– 03:00 hr, with nearly every single crayfish active during those times during the study (Fig 4). After 03:00 hr, crayfish activity slowly declines with surface activity rarely being observed at 17:00hr. Despite being less active throughout the day, 25% to 50% of time each crayfish remained active throughout the daytime.

### Environmental predictors of surface activity

*Lacunicambarus thomai* surface activity was best predicted by a model containing the effect of time, humidity, temperature, an interaction between time and humidity, and an interaction between humidity and temperature (S1 Table). However, the three most likely models were all within 2 AICc values of the best fit model (S1 Table). Based on the model averaged coefficients, humidity, temperature, and the interaction between humidity and temperature all predicted the activity of *L. thomai* throughout the duration of our study (S2 Table). However, the humidity variable had the strongest overall effect (S2 Table).

### Surface activity

Among the six crayfish observed, there was a high degree of surface level activity. Each crayfish spent at least 20% of the time that we filmed at their burrow portal. The percentage of total time each crayfish was observed at the surface ranged from 21% (9.18 hr) of the total filmed time to 69% (39.42 hr); with an average of 42% of the time spent active across all six crayfish

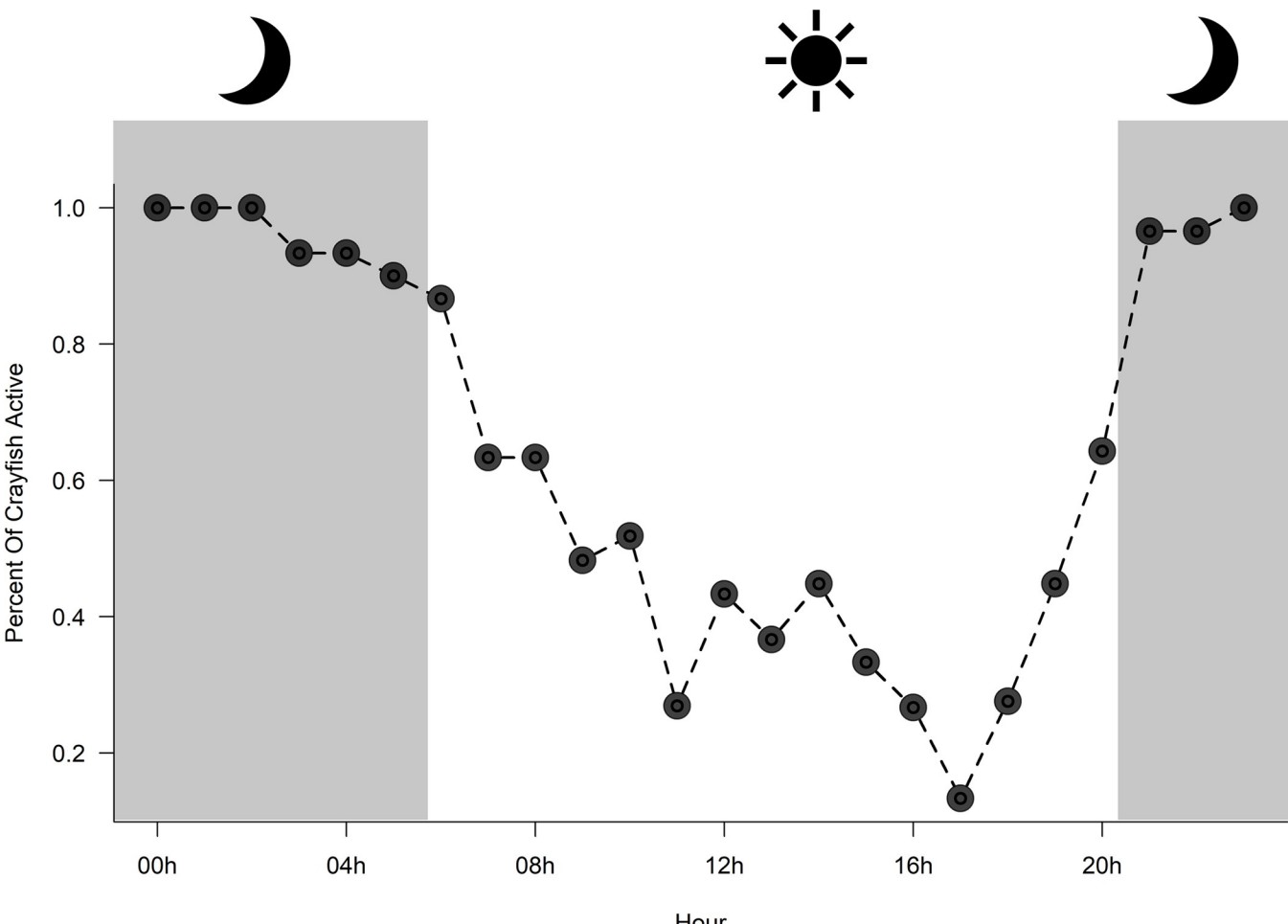

**Fig 4. The percent of *Lacunicambarus thomai* (n = 6) that were active for each hour during our study.** Gray areas represent the civil twilight time for our study period that represents the transition between day and night. Crayfish were more likely to be active during the night than during the day. Data is reported from all crayfish throughout the entire period of our study.

(Fig 5). Interestingly, one of our six crayfish (crayfish 6) spent a greater portion of the time active above ground than underground. The remaining 5 *L. thomai* spent a greater percentage of time underground, with a range of 31% (17.92 hr) to 79% (120.47 hr) of their time spent underground and with an average of 58% of each crayfish's time spent underground (Fig 5).

## Day vs. night surface activity

All *L. thomai* exhibited a high degree of surface activity both during the day and during the night. The percentage of time that crayfish were active during the daytime ranged from 20% (7.71 hr) to 73% (13.65 hr) of the time with an average of 36.46% of activity occurring at the burrow portal or on the surface (Fig 6). Regarding nighttime activity, five of the six crayfish exhibited more nighttime activity at the surface of their burrow than they did daytime activity. Night activity ranged from 27% (5.15 hr) to 80% (30.46 hr) with an average of 63% of the total surface activity occurring during the night (Fig 6).

The behaviors exhibited by *L. thomai* varied depending on whether they were active during the day or night (S3 Table). The most common diurnal behavior recorded was the guard and relax behavior, which comprised nearly 80% of the total time crayfish were observed at the

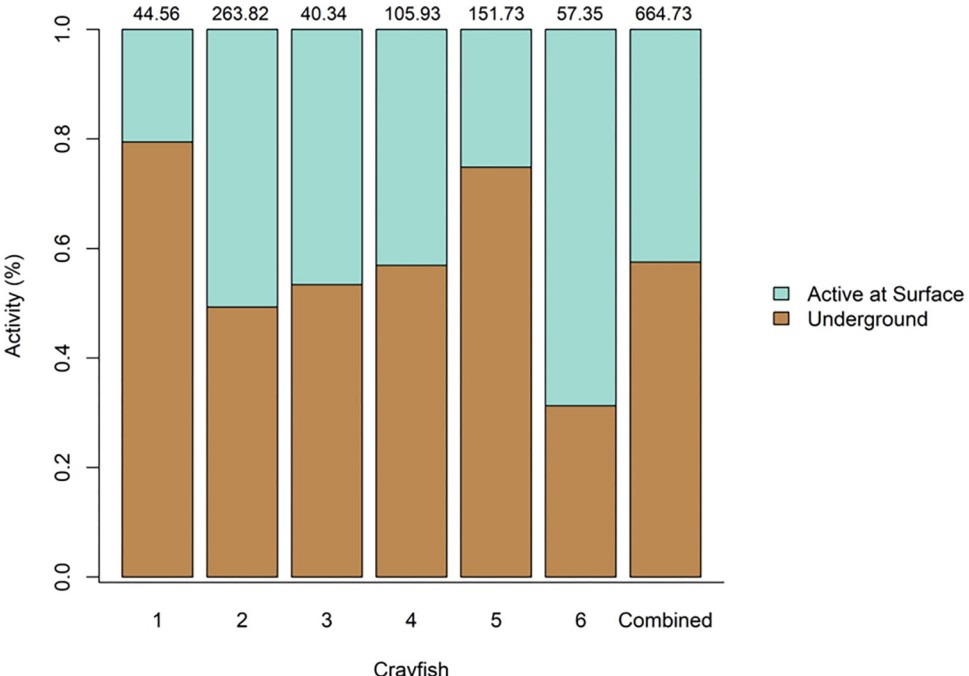

**Fig 5. The percentage of time that *Lacunicambarus thomai* spent visible at the surface of their burrow (active at surface) or not visible underground in their burrow (underground).** We report the percentage of time that each crayfish individual spent active or inactive as well as a combined total for all six crayfish. For each crayfish, the total number of time (hours) that each crayfish was filmed is listed above each bar.

surface during the day (Fig 7). Interestingly, a single crayfish (crayfish 4) exhibited foraging behavior during the day for 0.001% (392 s) of the time that they were filmed. Because this was such a small portion, it is not visible in Fig 7A. We observed two crayfish (crayfish 1 and 5) excavating their burrow during the day. Overall, each crayfish spent most of their day activity exhibiting the guard behavior at their burrow portal (Fig 3; S3 Table).

Crayfish showed a greater level of nocturnal surface activity. The majority of their time was spent in the relax position during the night, and relaxing was more likely to occur during the night compared to during the day (Fig 7); both of these behaviors occurred at burrow portals. Interestingly, each crayfish spent a portion of their nighttime activity exhibiting foraging behavior, with an average of 11% of all night activity including foraging (Fig 7). The same two crayfish that excavated during the day (crayfish 1 and crayfish 5) were also observed excavating their burrow at night, but they excavated for a greater percentage of their nighttime activity than their daytime activity (Fig 7).

When analyzing the likelihood of each behavior occurring at day versus night, the chi-squared analyses demonstrated that every behavior was more likely to occur during one light period compared to another. Specifically, the guard behavior and inactivity (i.e., under) were more likely to occur during the day compared to the night (Fig 8; S4 Table). By contrast, excavate, forage, and the relax behavior were more likely to occur at night (Fig 8; S4 Table). The hunt behavior was also more likely to occur during the night than during the day (Fig 8; S4 Table).

## Discussion

Out of the 664.73 hrs that we observed across six *L. thomai*; crayfish were active at the surface of their burrow nearly half of this time. Furthermore, we observed a surprisingly high amount

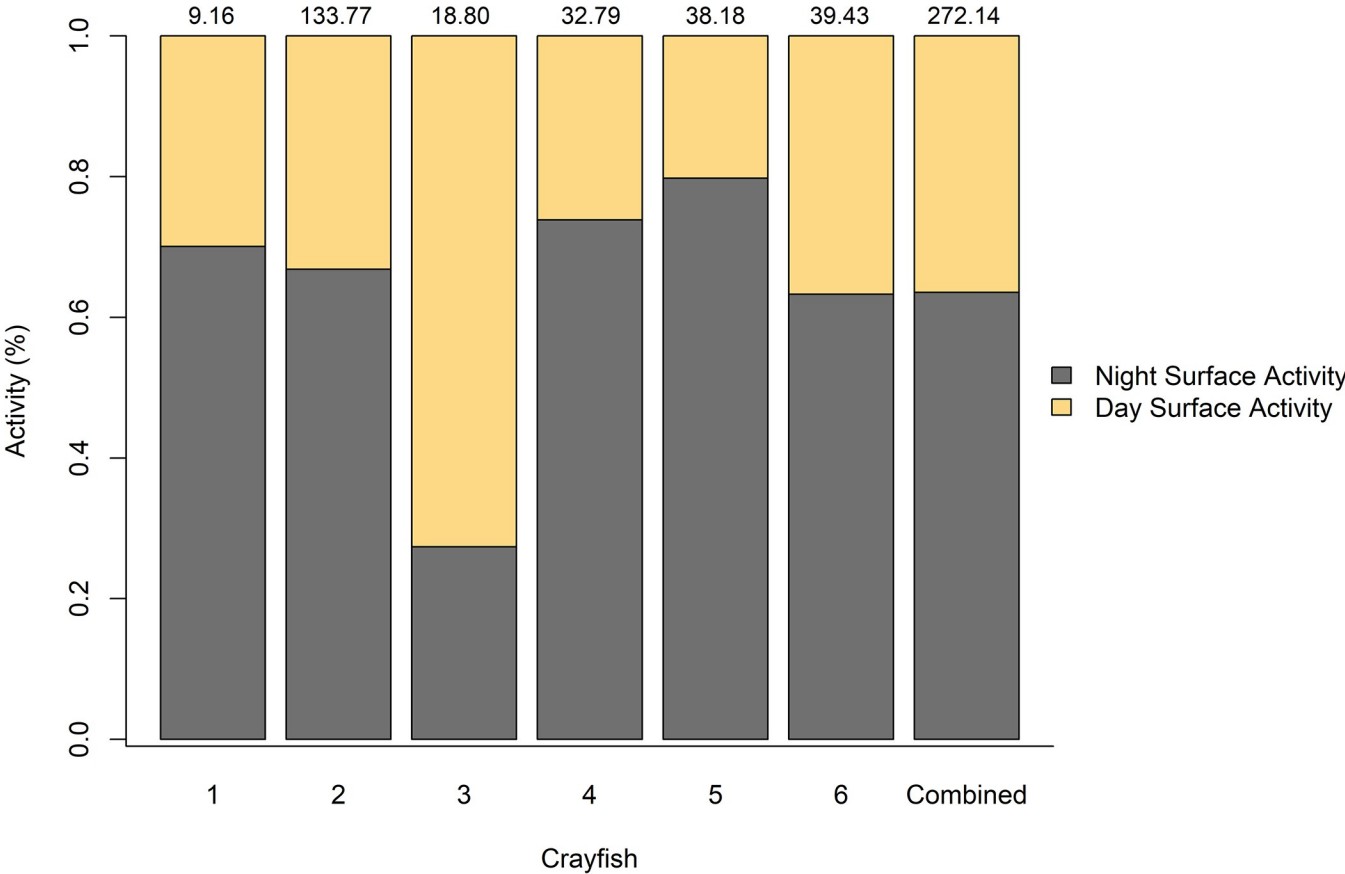

**Fig 6. The percentage of time that *Lacunicambarus thomai* was observed exhibiting surface activity during the day and during the night.** We report the percentage of time that each individual crayfish spent during the day and the night as a combined total for all six crayfish. For each crayfish, the total number of time (hours) that each crayfish was filmed active is listed above the bar.

of surface activity during the day, with nearly one-third of all surface activity occurring during the day and not at night. With our footage, we were also able to create an ethogram based on the five different surface behaviors (relaxed, guard, forage, excavate, and hunt) that we observed. Our ethogram allowed us to examine the behavioral repertoire and prevalence of the whether the observed behaviors more frequently occurred during the day or night. We discuss these findings in relation to previously published observations of burrowing crayfish surface behavior as well as potential opportunities for future research.

Before our study, little has been reported on the degree of surface activity and specific behaviors exhibited by burrowing crayfishes. Although surface activity has been observed throughout burrowing crayfishes both during the day and during the night, the prevalence in which *L. thomai* was active at the surface far exceeds that of other previously published reports. Previous accounts suggest that burrowing crayfish were only active above ground during a few instances which included high water events [51], during cloudy days according to light levels [36], or when looking for food or mates [12]. However, many of these accounts come from *in situ* natural history observations, and not systematic investigations. For example, Loughman et al. 2015 reports over 50 observations of *Cambarus pauleyi* at the surface of their burrow, but only three of these observations were made during the day [52]. Moreover, Foltz et al. 2018 states that *Cambarus loughmani* did not rise from burrows until a few hours after twilight, with no daylight activity reported [26].

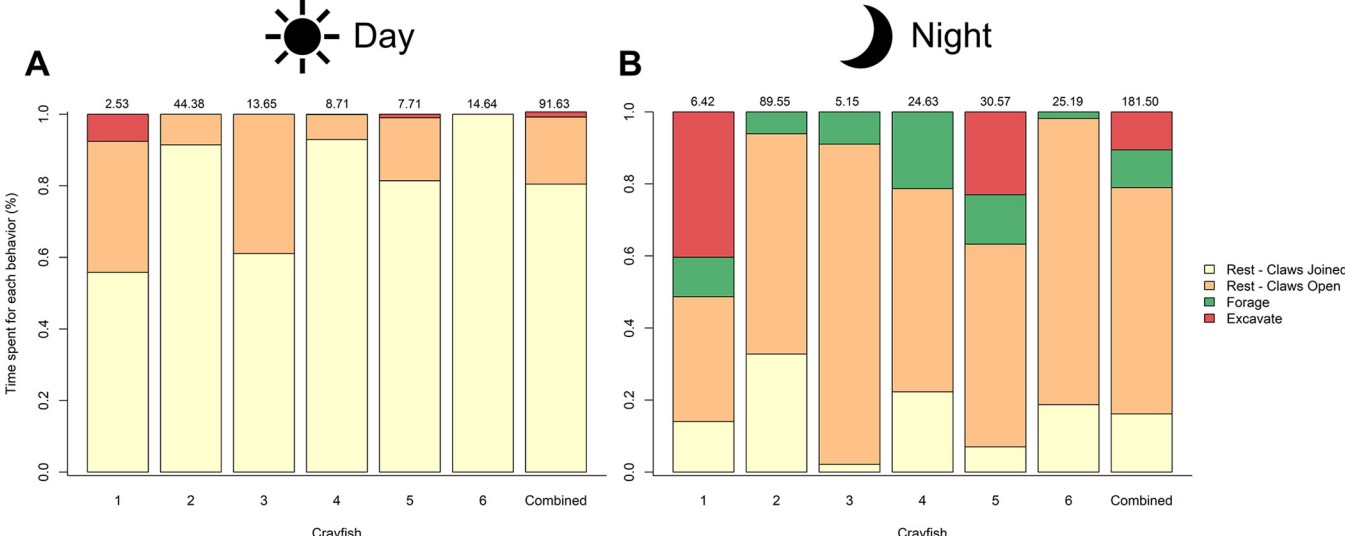

**Fig 7.** The percentage of time that each *Lacunicambarus thomai* spent exhibiting different behaviors throughout the (A) day and the (B) night. We report the percentage of time that each individual crayfish spent during the day and the night and combined six crayfish. The hunt behavior was not included in because of the relatively small time that this behavior was exhibited. For each crayfish, the total number of time (hours) that each crayfish was filmed active is listed above the bar.

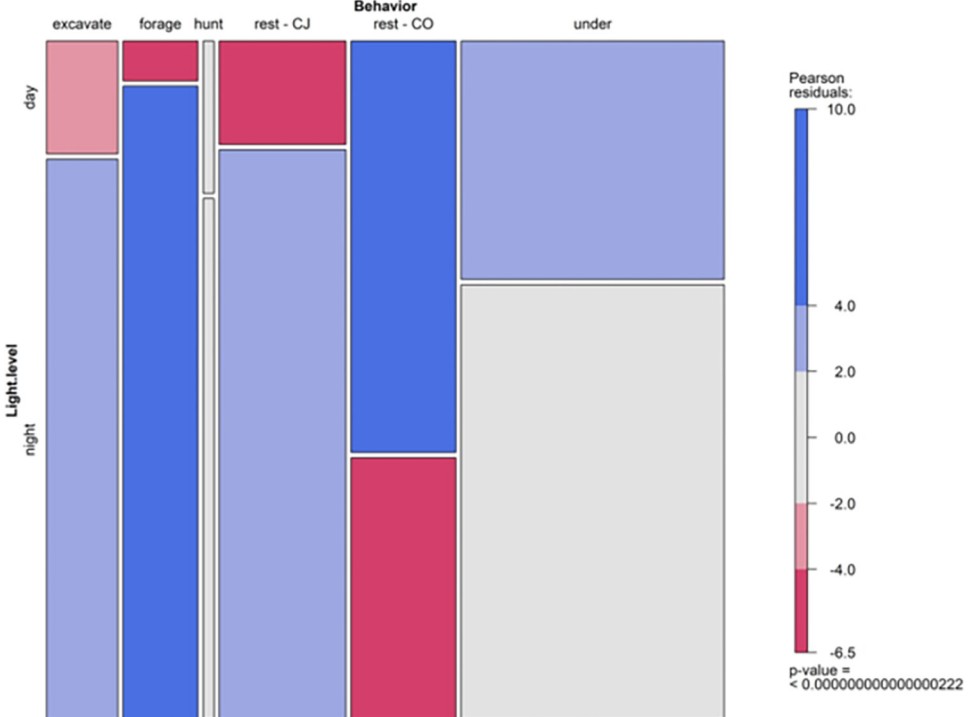

**Fig 8. Mosaic plot of the behavior exhibited by each crayfish (excavate, forage, guard, hunt, relax, under) based on the time of day it was recorded (night, day).** The width of the bars represents the number of observations that we recorded, whereas the height of the bars represents the proportion of observations that were recorded in the night (the bottom bar) versus the day (the top bar). The color of each bar represents whether the observed frequency of each behavior deviates from the expected frequencies if the variables were independent. Red denotes observed frequencies that are smaller than the expected frequency, whereas blue represents the observed frequencies that are larger than the expected frequencies. Gray represents a similar observed frequency when compared to the expected frequency.

Although we studied a small number of *L. thomai* during a two-month summer period, every crayfish was active during peak activity times. Only two previous studies have reported hourly activity data for burrowing crayfishes [35, 38]. For over a year, Bearden et al. (2021) collected hourly activity data for two burrowing crayfish species (*Lacunicambarus erythrodactylus* and *Procambarus holifieldi)*. In their study, both *L. erythrodactylus* and *P. holifieldi* were observed active roughly 25% of the time during the night and were rarely observed being active during daytime hours [38]. In a laboratory study, Palaoro et al. 2013 collected hourly data on a South American burrowing species (*Parastacus brasiliensis*) and showed that this species spent a majority of their experiment within their burrow, but were more likely to leave their burrow during the night compared to during the day [35].

In our study, the high degree of surface activity observed may be specific to crayfish in the genus *Lacunicambarus*. However, because our study was conducted on few individuals, future research on the surface behavior of *Lacunicambarus thomai* and the other 13 species in the genus *Lacunicambarus* is warranted. Alternatively, other burrowing species (and genera) may also be highly active during the day, but researchers often exclusively collect burrowing crayfish during the night which can bias the prevalence of these behavioral observations [24, 51]. For example, in a seven year-long ecological study of the Digger Crayfish, *Creaserinus fodiens*, Norrocky et al. 1991 reported that all crayfish collections occurred at night [51]. Clearly, the high degree of surface activity that we found in *L. thomai* contrasts our previous understanding of the activity and surface behaviors of burrowing crayfishes. Further, our study solely reports observations of surface behavior, and we are unable to report on the presence, absence, or degree of underground activity.

Based on our analysis of the environmental predictors of surface activity, we found that humidity had the strongest overall effect of predicting burrowing crayfish surface activity. Specifically, we found that crayfish were more likely to be active at lower humidity values. These results contrast previous findings by Bearden et al. 2021, in which activity was best predicted by a combination of variables, including daylength, air temperature, and groundwater temperature. Importantly, our analysis only uses reported data from the macro-environment at a nearby weather station, and not local, micro-environmental variables (variables recorded at the exact environment of or study site), as was done by Bearden and colleagues [38]. For example, in their study, Bearden et al. (2021) found that chimney constructed was related to precipitation events. It remains unknown whether the other surface behaviors which we describe have similar correlations to environmental variables such as prescription or groundwater temperature. Future studies should take into account the variation within environmental variables such as air temperature and burrow temperature to investigate behavioral patterns in burrowing crayfish. For example, activity in burrowing mammals is often best predicted by an interaction between both surface and groundwater temperatures [53]. Therefore, similar surface and groundwater temperature interactions may occur in burrowing crayfishes which are already known to use their burrows as refuges from harsh environmental temperatures [12, 13, 38].

Besides our study, only one other study has created an ethogram specific to burrowing crayfish surface behavior, although this study was conducted in a laboratory environment [35]. By contrast, our *in-situ* recordings allowed us to create an ethogram based entirely on natural behaviors, which revealed the richness of burrowing crayfish surface behavior—many of which have been previously undocumented. In our study, because *L. thomai* were constantly being monitored, we were able to categorize each crayfish into one of five different surface behaviors. The two most prevalent activities were the relax behavior and the guard behavior, which we separated based on the orientation of the crayfish's claws relative to their body position (Fig 3). Whether the guard behavior's function is truly to defend the burrow is unknown,

because we did not observe any intraspecific interactions in our study. However, defense, guarding-like behavior has been reported in other burrowing crayfish [51].

Interestingly, there were strong differences in the prevalence of the guard and relax behaviors depending on whether it was during the day or night. We found that during the day, *L. thomai* was more likely to be active in the guarded position compared to the relax position. By contrast, when we observed *L. thomai* active during the night, they were much more likely to be in the relax position than the guard position. The function of these different postural behaviors is unknown and requires further investigation. But we can speculate that the guard behavior is more likely to be exhibited during the day due to the increased potential for predator interactions. Indeed, we observed terrestrial predators (i.e., snakes and birds) interacting with *L. thomai* burrows during the day [37]. Alternatively, the rest behavior being more likely to occur during the night may relate to this posture's increased ability to quickly capture live prey that move near the burrow [36, 37, 42]. Future studies must investigate whether or not these postural differences are present throughout other burrowing crayfishes, or if they are specific to *L. thomai*.

We also observed 271 instances of *L. thomai* engaging in active foraging for nearby vegetation. Within our observations, *L. thomai* often explored the area within the vicinity of their burrow portal and used their claws to cut nearby vegetation. Across all our observations, only a single instance of vegetation foraging occurred during the daytime, which demonstrates a strong preference for this behavior occurring at night when there are presumably fewer active predators. Interestingly, the vegetation was either immediately consumed at the base of the burrow or was taken down into the burrow out of the view of the camera. Therefore, our observations confirmed previous reports of burrowing crayfish foraging; because the area surrounding the burrow portals are often replete with vegetation [26]. Furthermore, crayfish burrows are commonly excavated into large chambers, and large amounts of vegetation are found within the burrow tunnels and the burrow chambers, which are likely to serve as a food storage cache [26, 27, 54]. Although our observations confirm that *L. thomai* does take vegetation down into the burrow it is difficult to confirm the hypothesis that *L. thomai* burrows may serve as a food cache. The use of crayfish burrowing chambers in laboratory settings [30] is required to confirm the possibility for crayfish using their burrows to store and consume food resources.

In addition to observations of *L. thomai* foraging for vegetation, we observed 32 instances of the crayfish exhibiting sit and wait hunting behavior. In these observations, the crayfish sat at the burrow portal and quickly lunged at live animal prey (i.e., the hunt behavior). In some instances, this ambush behavior successfully resulted in the capture of live animal prey, whereas other attempts did not result in successful acquisition of the prey. When this behavior resulted in a successful attempt, the crayfish immediately took the item into their burrow. This ambush behavior was more likely to occur nocturnally than diurnally, with only three of 32 observations occurring during the day. Interestingly, this sit and wait predatory behavior seems to be prevalent throughout the genus *Lacunicambarus*, as it has been reported in at least 2 other *Lacunicambarus* species [36, 37, 42]. Recently, Thoma (2022) reports that in Ohio, *Lacunicambarus chimera* is never active at the surface of their borrow, whereas another species, *Lacunicambarus nebrascensis* rarely exhibits surface activity [55]. Although other species of crayfish have been observed capturing live animal prey [28, 29], the degree and reliance on this ambush predation behavior across burrowing crayfishes is unknown.

Across the six crayfish that we studied, we only observed two *L. thomai* actively excavating their burrows. These observations indicate burrow excavation is not an exceedingly common *L. thomai* behavior [35, 38], and may only occur when necessary alterations to the burrow architecture are needed. Each observation of excavation behavior always occurred in a

repeated series of ritualized behaviors. First, crayfish would be observed bringing mud up to the burrow portal. Once at the surface, the crayfish would immediately stop before reaching ground level and immediately tap their antennae at their edge of their burrow portal. Lastly, the crayfish would finally emerge and place the mud pellet and methodically use its claws the push the mud into position.

In previous studies of *P. holifieldi*, precipitation was one of the variables that influenced the likelihood of burrow excavation (i.e., referred to as chimney construction in the study) [38]. Indeed, burrows become flooded during times of rainfall and often require renovations following these events (Z. Loughman, personal observations). Only two small rain events occurred during our study period, which may explain the lack of burrow excavation being observed. Furthermore, excavation behavior was more likely to occur at night than during the day. Intuitively, burrow excavation is a relatively costly behavior, given substrates must repeatedly be excavated from the bottom of the burrow to the surface. Such behaviors are more likely to have increased costs during the day, both based on the high surface temperatures and the potential for increased rates of predation during the daytime.

In summary, based on our preliminary investigation, we found that *L. thomai* exhibits a high degree of surface activity. Further, we found that *L. thomai* exhibits several behaviors at the their burrow portal, and these behaviors all were more likely to occur during either day or night. However, our study brings several unanswered questions to light regarding the behavior of burrowing crayfishes. For example, although we observed many small foraging excursions that occurred within the immediate vicinity of the burrow, we did not observe a single large-scale excursion during this time. In other species of burrowing crayfishes, substantial surface activity occurs immediately after rainfall, and such crayfish may travel large distances in a short period of time [12; Z. Loughman & Z. Graham personal observations]. Although the exact function of these excursions is unknown, it is unclear whether or not *L. thomai* engages in such behaviors.

Further, throughout our study, we did not observe any intra-specific interactions. Again, scattered observations highlight the social behavior of burrowing crayfishes, but interestingly, we did not observe any social behavior throughout the course of our study. Future studies should conduct similar behavioral observations throughout the year and also investigate potential demographic differences in the behaviors we report. By expanding the time frame in which records occur, social interactions of larger foraging excursions may be recorded. Lastly, although our study has shed light on the activity of crayfishes at the surface, much remains unknown about the subsurface behavior of burrowing crayfishes. Additionally, since we discovered that this species displayed easily identifiable postural behaviors, this may allow for automated video analysis of body position, activity, and behaviors. As such, the continued use of burrowing crayfish observations chambers [30] will be paramount in furthering our understanding of the behavioral ecology of burrowing crayfishes. In conclusion, our study revealed a surprisingly high degree of surface activity in a species of burrowing crayfish. Based on the historical difficulties in studying burrowing crayfish, there is much to learn about the ecological and evolutionary dynamics of burrowing crayfish behavior.

## Supporting information

**S1 File. Recording of an excavation observation during the night.**
(MP4)

**S2 File. Recording of a foraging observation during the night.**
(MP4)

**S3 File. Recording of a hunt observation during the night.**
(MP4)

**S1 Table. The most likely model predicting the activity of burrowing crayfish based on time, humidity, temperature, an interaction between time and humidity, and an interaction between humidity and temperature.**
(DOCX)

**S2 Table. Coefficients and standard errors (SE) for the model predicting the activity of *L. thomai* based on full model averaging.**
(DOCX)

**S3 Table. The number of observations and the mean duration of each behavior observed during our study.**
(DOCX)

**S4 Table. Results from Chi-squared test for the number of behaviors that occur during the day and versus the night.**
(DOCX)

**S1 Fig. Average, minimum, and maximum air temperature at our study location in Hickory, PA throughout our study period (June 2020 –July 2020).**
(PNG)

**S2 Fig. Average, minimum, and maximum humidity at our study location in Hickory, PA throughout our study period (June 2020 –July 2020).**
(PNG)

**S3 Fig. Average, minimum, and maximum air pressure at our study location in Hickory, PA throughout our study period (June 2020 –July 2020).**
(PNG)

**S4 Fig. Precipitation at our study location in Hickory, PA throughout our study period (June 2020 –July 2020).**
(PNG)

## Acknowledgments

We thank the Davis family for the use of their land for the study location.

## Author Contributions

**Conceptualization:** Kaine M. Diehl, Destinee A. Davis, Zachary J. Loughman, Zackary A. Graham.

**Data curation:** Kaine M. Diehl, Nicoleena M. Storer, Hogan D. Wells, Destinee A. Davis, Zackary A. Graham.

**Formal analysis:** Zackary A. Graham.

**Investigation:** Kaine M. Diehl.

**Methodology:** Kaine M. Diehl, Destinee A. Davis.

**Supervision:** Zachary J. Loughman, Zackary A. Graham.

**Writing – original draft:** Kaine M. Diehl, Nicoleena M. Storer, Hogan D. Wells, Destinee A. Davis, Zachary J. Loughman, Zackary A. Graham.

**Writing – review & editing:** Kaine M. Diehl, Nicoleena M. Storer, Hogan D. Wells, Destinee A. Davis, Zachary J. Loughman, Zackary A. Graham.

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
