## [Decision Letter · Decision Letter 0]

3 May 2022

PONE-D-21-39709On the surface or down below: Field observations reveal a high degree of surface activity in a burrowing crayfish, the Little Brown Mudbug (Lacunicambarus thomai)PLOS ONE

Dear Dr. Kaine,

Thank you for submitting your manuscript to PLOS ONE. After careful consideration, we feel that it has merit but does not fully meet PLOS ONE’s publication criteria as it currently stands. Therefore, we invite you to submit a revised version of the manuscript that addresses the points raised during the review process.

We look forward to receiving your revised manuscript.

Kind regards,

Junhu Dai, Ph. D.

Academic Editor

PLOS ONE

Journal Requirements:

Additional Editor Comments:

Please modify the manuscript by strictly abiding by the reviewers' comments and their suggestions. Since one of them made a reject conclusion, you should be more careful for the revision of this time. Good luck to you.

Reviewers' comments:

Reviewer's Responses to Questions

**Comments to the Author**

1. Is the manuscript technically sound, and do the data support the conclusions?

Reviewer #1: Yes

Reviewer #2: Partly

2. Has the statistical analysis been performed appropriately and rigorously? 

Reviewer #1: Yes

Reviewer #2: No

3. Have the authors made all data underlying the findings in their manuscript fully available?

Reviewer #1: Yes

Reviewer #2: Yes

4. Is the manuscript presented in an intelligible fashion and written in standard English?

Reviewer #1: Yes

Reviewer #2: Yes

5. Review Comments to the Author

Reviewer #1: The authors made an interesting work, continuously filming the surface behavior of crayfish which were thought before to stay mostly or totally in underground burrows. They registered activity both during the day and night along with daily environmental factor variations and analyzed their behavior through the video recordings. While most behavioral studies in crayfish have been done in laboratory settings, this one contributes by rescuing ecological context and opening new perspectives of research. It is mostly well done but there are three general issues to be considered:

1) One of the big contributions of this work is to enable discrimination of individual behavior, which is not a simple task in natural observations. Individual data presentations are fine but I suspect that when average analysis are presented, some or most of them are not considering that one individual (#2) with more than double the number of data collection is biasing the results. This can be deduced from some graphs while others do not let us know because description of average calculations are missing in Materials and Methods.

2) While behavioral analysis can be made when these crayfish are on surface, nothing can be stated of their underground times. In this sense, it is an error to assign “inactivity” for underground times.

3) Discussion needs to be improved.

Comments:

- Study Site: More basic description is required of the study area around the filming spot, to better explore the results. It is said that most species (tertiary and secondary burrowing crayfish) inhabit lotic and lentic environments but it is not clear how far are the Primary burrowing species of this study from water bodies, for instance. It is said in the Introduction that “Lacunicambarus thomai is a burrowing crayfish species with a high propensity to inhabit burrows in marshes, roadside ditches, and flooded fields (38). Populations of L. thomai often live in localized colonies with conspecifics and inhabit burrows that are relatively simple but can nonetheless be up to 1-1.5 m deep (38,39)”. However, nothing, no information is provided of the specific area where the study took place: proximity from rivers, vegetation types and cover densities, average ranges of daily environmental factor variations, Latitudinal coordinates.

- Burrows were selected and filmed. How far is one burrow from the other? Inform the average land area inside which the 6 burrows were located. One individual is associated to each burrow – is that an assumption or are there evidences that they are solitary? Furthermore, it is said that burrows may have more than one entrance (line 67), how was this issue treated here?

- It is said in Statistical Analysis that none of the independent variables (humidity, temperature, and time) were strongly correlated (all r < 0.6). A representative graph depicting an average 24h variation of environmental factors in the studied season would be informative in Supplemental Material. How was time included in the models, as a categorical or continuous variable? A continuous variable with linear increase (such as a sequence from 0 to 23) would create artificial associations in the model. Please evaluate, based on the statistical parameters found in Table 3, if it is not enough to use a simple model without interaction between time, humidity and temperature to understand the influence of environmental factors on the surface activity. The reason for this question is that the complexity of the best-fit model seems to have inhibited any discussion about the analysis in the end.

- The Results section start with “Hourly Activity” (which should be “Hourly surface activity”) and Figure 4, but there is no explanation as to how this was calculated. It is explained in Statistical Analysis that for an individual to be considered active on surface in one specific hour and day, it needs to be seen in any time point within that hour in that day. Then, how was the group average/percentage calculated taking into account that each contributing individual was registered for a different number of days? Individuals that had more filmed days should not weight more. Finally, Legend Figure 4 needs to inform that averages were calculated taking into account all individuals and all days.

An explicit description of calculation should also be added to “percentage of time spent active throughout daytime” and “ proportion of time spent on surface”. In Table 5, how was the “mean duration” of each behavior calculated, taking into account individuals and number of days each individual was filmed?

In Figures 5, 6 and 7, it is interesting to show the “combined” proportions. However, it is biased by individual 2, which was filmed for more days. Could an unbiased calculation be made here?

In Figure 8 the number of observations is again biased to individual 2.

- Figure 5 and associated text: comparison should be between “surface” and “underground”, not between “active” and inactive” because nothing is known about what the crayfish are doing underground. The same for Figure 6: “nighttime on surface” and “daytime on surface”. In Line 281, replace “ active” by “on surface” in “The percentage of time that crayfish were active during the daytime”. In Line 283, “Regarding nighttime activity on surface”. Legend Figure 6: “percentage of time that each individual crayfish spent on surface during the day and the night”.

-

- Discussion: In contrast to Bearden et al. (2021), this study brings more information about the behavioral complexity of this particular crayfish species. However, the lengthy discussion is mainly descriptive of results. The authors should explore, for instance, what was found in statistical analysis and how could this be connected to the specificity of the studied environment, to take full advantage of the in situ study. Another suggestion is to take Bearden et al. (2021) as a reference, discuss how the results are constrained by the particular season and microhabitat that was covered in this study.

- Humidity was indicated as the most important factor modulating surface appearance in crayfish. This variable, as well as all others were collected from a meteorological station. Any thoughts about the validity of using only macro-environmental measurements in association with behaviors that are restricted to the spatial scale of the entrance of a burrow?

- Temperature was shown to be a strong predictor of surface activity. A strong suggestion for future studies is to also consider underground temperature in this analysis. For instance, it has been shown in endothermic subterranean rodents that a combination of external and underground temperatures predict better the episodes of surface emergences (Jannetti et al., Conservation Physiology 2019 v. 7, p. coz044). It is reasonable to assume that similar influence is potentially valid for these crayfish.

- This crayfish display clear postural signatures that enable behavioral identification through the relative position of body coordinates (Figure 3). This could potentially be used in future automated video analysis of activity patterns.

Minor Comments:

Fig.2C: view of the camera

Line 248: remove extra “were”

Line 319: “every single behavior was more likely to occur during day or night compared to another” An alternative could be “daily phase”

Table 4: what does this mean? “the activity of L. thomai was negatively related to the activity of crayfish.”

Reviewer #2: I understand that behaviour of strictly burrowing crayfish is difficult to study, so the information presented by the authors is surely interesting, and novel. However, the entire manuscript is based on only six individuals that were not even characterized. So, data are very preliminary and should be interpreted more cautiously. Behavioural categories should be also analysed together, considering that are dependent data. I am sorry not to be more positive, but a more specialised journal seems to be more appropriate for the manuscript.

Lines 50-52, 109: for the readers it would be interesting to specify that these species are North American

Line 67: maybe opening is more suitable than portal

Lined 128-130: so how many burrows were checked before selecting only six? How about the density of the burrows in the study area?

Line 132: crayfish could have been attracted out of the burrows with some baits after the footage to characterize them (for hunt behaviour it is reported they leave the burrow for example)

Line 182: were the behavioural data checked for normality?

Lines 218-222: those behaviours are dependent each other, so it is better to analyse them together because when crayfish are guarding, for example, they are not feeding or hunting

Line 235: please delete during the study

Line 270: please 5 not in italics

I suggest merging Fig 5 and 6 in one figure

Table 5: please report all the duration in seconds. Moreover, please change during with duration in the caption

Tables should be better drafted and presented

Line 343: please correct five

Line 345: please better rephrase this sentence

Line 356: Loughman et al. 2018 is not present in the bibliography; 46 is Loughman et al. 2015

Line 371, 448: please consider that only six individuals were observed, so I suggest being more cautious in this statement

Line 381: please do not use intricacies but richness or diversity

Line 415: please correct “is required”

Line 436: I think it is “use its claws to push the mud”

6. PLOS authors have the option to publish the peer review history of their article (what does this mean?). If published, this will include your full peer review and any attached files.

Reviewer #1: No

Reviewer #2: No

---

## [Author Response · Author response to Decision Letter 0]

18 May 2022

Dear Dr. Junhu Dai,

Thank you for allowing us the opportunity to revise our manuscript. The comments from the two reviewers were extremely helpful and made the manuscript much stronger. Based on their suggestions, we incorporated most suggestions to our manuscript. 

Below, you will find a letter detailing how we dealt with each comment of each reviewer. Where applicable, we state what we have changed and the location in which you will find our modifications in our updated manuscript. 

We address each reviewer comment (C#) in individual answers (A#) to keep everything organized.

Reviewer #1: 

C1: The authors made an interesting work, continuously filming the surface behavior of crayfish which were thought before to stay mostly or totally in underground burrows. They registered activity both during the day and night along with daily environmental factor variations and analyzed their behavior through the video recordings. While most behavioral studies in crayfish have been done in laboratory settings, this one contributes by rescuing ecological context and opening new perspectives of research. It is mostly well done but there are three general issues to be considered:

A1: Thank you for the generous comments. We appreciate your suggestions and we have attempted to address all your comments below. 

C2: One of the big contributions of this work is to enable discrimination of individual behavior, which is not a simple task in natural observations. Individual data presentations are fine but I suspect that when average analysis are presented, some or most of them are not considering that one individual (#2) with more than double the number of data collection is biasing the results. This can be deduced from some graphs while others do not let us know because description of average calculations are missing in Materials and Methods.

A2: We agree that analyzing behavioral data presents several issues and that this issue can be exacerbated by having an unequal sample size (263 hrs of footage for crayfish 2 vs 40 hours for crayfish 1). This is why we have chosen to present the majority of our results in terms of the percentages and not raw values. Reporting this information in terms of the raw values would certainly bias our results, as you suggest. However, by reporting this information (Figures 5, 6, and 7) in percentages, we believe that this is the most appropriate way to report our results and adjust for the bias. Furthermore, our manuscript reports relatively few true statistical analyses (and instead opts for more descriptive statistics of what we observed). This aligns with the primary goals of our manuscript—to describe the behavioral diversity of this elusive crayfishes and demonstrate how this new methodology can be lucrative in the field of crustacean behavioral ecology. 

C3: While behavioral analysis can be made when these crayfish are on surface, nothing can be stated of their underground times. In this sense, it is an error to assign “inactivity” for underground times.

A3: We agree. Our manuscript can only report of the surface activity, and not any potential underground behaviors. We have altered our language throughout the entire manuscript to account for this comment. We have changed any discussion of “inactivity” to “inactivity at the surface”. 

C4: Discussion needs to be improved.

A4: We have taken several of your comments regarding topics that require additional discussion and expanded on them. 

C5: Study Site: More basic description is required of the study area around the filming spot, to better explore the results. It is said that most species (tertiary and secondary burrowing crayfish) inhabit lotic and lentic environments but it is not clear how far are the Primary burrowing species of this study from water bodies, for instance. It is said in the Introduction that “Lacunicambarus thomai is a burrowing crayfish species with a high propensity to inhabit burrows in marshes, roadside ditches, and flooded fields (38). Populations of L. thomai often live in localized colonies with conspecifics and inhabit burrows that are relatively simple but can nonetheless be up to 1-1.5 m deep (38,39)”. However, nothing, no information is provided of the specific area where the study took place: proximity from rivers, vegetation types and cover densities, average ranges of daily environmental factor variations, Latitudinal coordinates.

A5: We agree that information on our study site was lacking in the original version of the manuscript. We have now included a detailed description of the location in which we conducted this study (see Lines 127-134). Because our sampling location is on a residential property, we choose to not report exact coordinates. 

C6: Burrows were selected and filmed. How far is one burrow from the other? Inform the average land area inside which the 6 burrows were located. One individual is associated to each burrow – is that an assumption or are there evidences that they are solitary? Furthermore, it is said that burrows may have more than one entrance (Line 67), how was this issue treated here?

A6: We have updated this information within our manuscript. We now provide information regarding how far away one burrow was from one another (Lines 131-132), as well as our assumption that each burrow was occupied by a single crayfish (Lines 141-144). We also clarify how we dealt with the issue with burrows with more than a single entrance (Lines 151-153).

C7: It is said in Statistical Analysis that none of the independent variables (humidity, temperature, and time) were strongly correlated (all r < 0.6). A representative graph depicting an average 24h variation of environmental factors in the studied season would be informative in Supplemental Material. How was time included in the models, as a categorical or continuous variable? A continuous variable with linear increase (such as a sequence from 0 to 23) would create artificial associations in the model. Please evaluate, based on the statistical parameters found in Table 3, if it is not enough to use a simple model without interaction between time, humidity and temperature to understand the influence of environmental factors on the surface activity. The reason for this question is that the complexity of the best-fit model seems to have inhibited any discussion about the analysis in the end.

A7: We have now included 4 new figures in the supplemental materials (Figures S1, S2, S3, and S4) which display the relevant environmental data during our study period. 

In the original manuscript, we reported that there were no strong statistical correlations between humidity, temperature, and time (as you mentioned). In our environmental analysis, time was coded as a continuous variable. We tested for the correlations beforehand to potentially deal with any artificial associations that you hint at. However, because of the low correlation values, we decided to proceed with using time as a continuous variable. Furthermore, the fact that time is not a strong predictor of activity (Table 4) suggests that there is no underlying association between these variables.

Our best fit model was quite complex, as you note. It was a model with 3 single parameters, and two interaction parameters. This why is we not only conducted a model-selection procedure (results in Table 3), but also the multi-model average procedure (results in Table 4). The results from our full model averaging technique allow us to evaluate all of the models together and construct effect sizes and weights for each of the terms in our model. This is why we come to the conclusion that humidity had the strongest overall effect (Table 4) compared to the other models. We have now included a better discussion of this result in our discussion section (see A14).

C8: The Results section start with “Hourly Activity” (which should be “Hourly surface activity”) and Figure 4, but there is no explanation as to how this was calculated. It is explained in Statistical Analysis that for an individual to be considered active on surface in one specific hour and day, it needs to be seen in any time point within that hour in that day. Then, how was the group average/percentage calculated taking into account that each contributing individual was registered for a different number of days? Individuals that had more filmed days should not weight more. Finally, Legend Figure 4 needs to inform that averages were calculated taking into account all individuals and all days.

A8: We have changed this section to “Hourly Surface Activity” as you suggest. 

We have clarified how this was calculated in the methods sections (Lines 201-209). Again, we report these values in terms of percentages, and not the raw number of hours each crayfish was active to avoid any potential sampling biases as you suggest. 

We have also updated the legend for Figure 4 to include those averages were calculated by taking into account for all individuals and all days throughout our study. 

C9: An explicit description of calculation should also be added to “percentage of time spent active throughout daytime” and “ proportion of time spent on surface”. In Table 5, how was the “mean duration” of each behavior calculated, taking into account individuals and number of days each individual was filmed?

A9: We have updated our methods to also include information as to how we calculated these percentages. By reporting these values in percentages, we avoid any issues based on the number of hours each crayfish was sampled and filmed. 

We have updated the legend in figure 5 to describe that our mean durations were calculated based on all crayfish. This data is reported to be entirely descriptive in nature, and therefore we believe that reporting such means are the most informative way to describe these different behaviors. 

C10: In Figures 5, 6 and 7, it is interesting to show the “combined” proportions. However, it is biased by individual 2, which was filmed for more days. Could an unbiased calculation be made here?

A10: Reporting our results in terms of percentages is the best way to provide an unbiased calculation for these figures, which are primarily meant to be descriptive and exploratory in nature. If we would have reported the data from the raw amount of time that each behavior/activity was performed, then this would certainly be biased. This is why we chose to reported the data in terms of percentages throughout the manuscript. 

C11: In Figure 8 the number of observations is again biased to individual 2.

A11: Yes, Figure 8 (but not 5, 6, and 7) is biased based on the increased number of observations from individual 2. This figure (and the analysis in general) are describing trends in whether or not a specific behavior was more or less likely to occur at day versus night. We have included information regarding this issue in our manuscript (Lines 201-203). Anecdotally, if you look at the percentages of all behaviors reported in Fig 5, 6, and 7, there is a similar degree of surface activity/behaviors being exhibited by each crayfish, which implies that this bias may be minimally impacting our results. 

C12: Figure 5 and associated text: comparison should be between “surface” and “underground”, not between “active” and inactive” because nothing is known about what the crayfish are doing underground. The same for Figure 6: “nighttime on surface” and “daytime on surface”. In Line 281, replace “ active” by “on surface” in “The percentage of time that crayfish were active during the daytime”. In Line 283, “Regarding nighttime activity on surface”. Legend Figure 6: “percentage of time that each individual crayfish spent on surface during the day and the night”.

A12: You are correct; we do not want to mislead the readers. Now, all of our figures have been accounted for the fact that we are only reporting surface activity and that we cannot comment on underground. Activity. Figures 5 and 6 are now updated accordingly. We have also changed the legend for Figure 6

C13: Discussion: In contrast to Bearden et al. (2021), this study brings more information about the behavioral complexity of this particular crayfish species. However, the lengthy discussion is mainly descriptive of results. The authors should explore, for instance, what was found in statistical analysis and how could this be connected to the specificity of the studied environment, to take full advantage of the in situ study. Another suggestion is to take Bearden et al. (2021) as a reference, discuss how the results are constrained by the particular season and microhabitat that was covered in this study.

A13: We agree that we could have expanded on our discussion section. Based on many of your suggestions below we believe that the discussion has significantly improved. 

C14: Humidity was indicated as the most important factor modulating surface appearance in crayfish. This variable, as well as all others were collected from a meteorological station. Any thoughts about the validity of using only macro-environmental measurements in association with behaviors that are restricted to the spatial scale of the entrance of a burrow?

A14: We agree that the macro environmental variables that we relate to surface activity are far from ideal for such studies. We have updated our discussion to point this out and suggest that future studies take into account the variables potential micro-environmental influences within and around the burrow entrances (Lines 406-422). 

C15: Temperature was shown to be a strong predictor of surface activity. A strong suggestion for future studies is to also consider underground temperature in this analysis. For instance, it has been shown in endothermic subterranean rodents that a combination of external and underground temperatures predict better the episodes of surface emergences (Jannetti et al., Conservation Physiology 2019 v. 7, p. coz044). It is reasonable to assume that similar influence is potentially valid for these crayfish.

A15: This is a great suggestion to add to our discussion section. We have now included that looking at the underground temperature variation is a fruitful area for future directions (Lines 416-422). 

C16: This crayfish display clear postural signatures that enable behavioral identification through the relative position of body coordinates (Figure 3). This could potentially be used in future automated video analysis of activity patterns.

A16: That is a great suggestion to add the discussion. We have included these postural changes to be used in the identification of automated video analysis in the future (Lines 511-513). 

C17: Fig.2C: view of the camera

A17: We have edited this sentence accordingly.

C18: Line 248: remove extra “were”

A18: We have removed the extra “were”

C19: Line 319: “every single behavior was more likely to occur during day or night compared to another” An alternative could be “daily phase”

A19: We believe that the current wording provides for clarity because “daily phase” may be confused with “day”. 

C20: Table 4: what does this mean? “the activity of L. thomai was negatively related to the activity of 

crayfish.”

A20: This was a typo and should have read “Thus, the activity of L. thomai was negatively related to the degree of environmental humidity.” We have made this change to the Table 4 text. Thank you.

Reviewer #2: 

C21: I understand that behaviour of strictly burrowing crayfish is difficult to study, so the information presented by the authors is surely interesting, and novel. However, the entire manuscript is based on only six individuals that were not even characterized. So, data are very preliminary and should be interpreted more cautiously. Behavioural categories should be also analysed together, considering that are dependent data. I am sorry not to be more positive, but a more specialised journal seems to be more appropriate for the manuscript.

A21: We agree that the fact that our study being only conducted on six individuals is a limitation to our study. Low sample sizes are typical for this type of work, because of the time intensive nature and complexities of naturalistic studies (see 6 individuals reported in Janettii et al. 2019) Although this work was only conducted on six individuals (that have unknown demographic information), we believe that the amount and nature of our data is worthy of publication. 

In the previous and updated versions of our manuscript, we attempt not to overstate our results as they are preliminary and exploratory in nature. Furthermore, although we do not report the demographic data from these crayfishes (sex, etc), there is no data that has alluded to sex differences in burrowing crayfish behavior. We do agree that this is an interesting and important angle for future studies though, so we have included that this information should be investigated in the future (Lines 507-508).

Despite these limitations, we still believe that we provide a novel methodology and interesting results (which reviewer 1 highlights) that will be of interest to a wide audience. Because the scope of PLOS ONE is to publish papers based on their scientific validity and methodology, we believe that our study is of interest to an audience outside of smaller, taxonomic focused journals.

We respond to the comment on behavioral categories being analyzed together below (see A27).

C22: Lines 50-52, 109: for the readers it would be interesting to specify that these species are North American

A22: We have edited Lines 50-52 to clarify that these are North American species (Line 50-51) 

C23: Line 67: maybe opening is more suitable than portal

A23: Portal is a term burrowed from the literature on mammalian burrowing behavior and is widely used to refer to crayfish burrow openings. Therefore, we prefer to keep this language consistent with prior published papers. We have included a few examples below. 

Glon, M. G., Adams, S. B., Loughman, Z. J., Myers, G. A., Taylor, C. A., & Schuster, G. A. (2020). Two new species of burrowing crayfish in the genus Lacunicambarus (Decapoda: Cambaridae) from Alabama and Mississippi. Zootaxa, 4802(3), 401-439.

Loughman, Z. J. (2010). Ecology of Cambarus dubius (upland burrowing crayfish) in north-central West Virginia. Southeastern Naturalist, 9(sp3), 217-230.

C24: Lined 128-130: so how many burrows were checked before selecting only six? How about the density of the burrows in the study area?

A24: We have now included much more information regarding our study location. Based on the relatively small number of burrows at the study location, we choose to only focus on a small sample size but to collect as much data as possible on each of these adult individuals. 

C25: Line 132: crayfish could have been attracted out of the burrows with some baits after the footage to characterize them (for hunt behaviour it is reported they leave the burrow for example)

A25: Yes, that is true. Unfortunately, we did not capture these individuals. We have included information how future studies need to explore how different demographics may exhibit such behaviors differently (lines 50-51). 

C26: Line 182: were the behavioural data checked for normality?

A26: Yes, all data and model fits were checked for normality. We have now included this information in the manuscript (Lines 197-199).

C27: Lines 218-222: those behaviours are dependent each other, so it is better to analyse them together because when crayfish are guarding, for example, they are not feeding or hunting

A27: This comment highlights the complexities of working with behavioral data, because an organism cannot perform more than a single behavior at once. Therefore, this is why we chose to primarily focus on broad, descriptive statistics throughout our study. Because of the novelty of these findings and the potential impact on our understanding of crustacean behavioral ecology, we believe that this is the proper way to analyze report our data at this stage. 

C28: Line 235: please delete during the study

A28: We have deleted this phrase.

C29: Line 270: please 5 not in italics

A29: We have unitalicized this number.

C30: I suggest merging Fig 5 and 6 in one figure

A30: We prefer to keep these figures separate as to avoid confusion between the messages of figure 5 (observed at surface vs. underground) and figure 6 (observed at surface during the day versus observed at surface during the night).

C31: Table 5: please report all the duration in seconds. Moreover, please change during with duration in the caption

A31: We have changed the table to report each behavior in. We have changed during to duration as you suggested. 

C32: Tables should be better drafted and presented

A32: Without any comments on the issues of the tables, we are unsure what to change base. 

C33: Line 343: please correct five

A33: We change made this change. 

C34: Line 345: please better rephrase this sentence

A34: We have rephrased this sentence accordingly (Lines 366-368).

C35: Line 356: Loughman et al. 2018 is not present in the bibliography; 46 is Loughman et al. 2015

A35: Thank you for noticing this issue. We meant Loughman et al. 2015 and we have made this change. 

C36: Line 371, 448: please consider that only six individuals were observed, so I suggest being more cautious in this statement

A36: We agree that we need to be more cautious with these statements. We have added additional information to these sections based on your comments (see Lines 394-395 and Lines 494-495).

C37: Line 381: please do not use intricacies but richness or diversity

A37: We have changed the work intricacies to richness as you have suggested. 

C38: Line 415: please correct “is required”

A38: We have changed “are required” to “is required”

C39: Line 436: I think it is “use its claws to push the mud”

A39: Thank you for catching this error. We have fixed the typo.

---

## [Decision Letter · Decision Letter 1]

19 Jul 2022

PONE-D-21-39709R1On the surface or down below: Field observations reveal a high degree of surface activity in a burrowing crayfish, the Little Brown Mudbug (Lacunicambarus thomai)PLOS ONE

Dear Dr. Diehl,

Thank you for submitting your manuscript to PLOS ONE. After careful consideration, we feel that it has merit but does not fully meet PLOS ONE’s publication criteria as it currently stands. Therefore, we invite you to submit a revised version of the manuscript that addresses the points raised during the review process.

Since I had difficulties finding a second Reviewer, I decided to review your revised manuscript myself. As you will see, the negative Reviewer of your first submission appreciated your corrections and judged that you contribution is now ready for publication. However, I found some problems that need to be settled before your paper is accepted. The main point concerns difficulties to link the results given in the text with those in the figures. I think that there may be some mistakes. For this reason, please carefully check your Result section. Also, I made several editorial recommendations. You will find all this information in the attached file "D-21-39709_R1_LFB.pdf". I appreciated your work and think that it will be a useful contribution to crayfish biology. 

We look forward to receiving your revised manuscript.

Kind regards,

Louis-Felix Bersier, Ph.D.

Academic Editor

PLOS ONE

Journal Requirements:

Additional Editor Comments:

See attached file "D-21-39709_R1_LFB.pdf"

Reviewers' comments:

Reviewer's Responses to Questions

**Comments to the Author**

1. If the authors have adequately addressed your comments raised in a previous round of review and you feel that this manuscript is now acceptable for publication, you may indicate that here to bypass the “Comments to the Author” section, enter your conflict of interest statement in the “Confidential to Editor” section, and submit your "Accept" recommendation.

Reviewer #2: All comments have been addressed

2. Is the manuscript technically sound, and do the data support the conclusions?

Reviewer #2: Yes

3. Has the statistical analysis been performed appropriately and rigorously? 

Reviewer #2: Yes

4. Have the authors made all data underlying the findings in their manuscript fully available?

Reviewer #2: Yes

5. Is the manuscript presented in an intelligible fashion and written in standard English?

Reviewer #2: Yes

6. Review Comments to the Author

Reviewer #2: I appreciate the responses and corrections provided by the authors, I do not have further comments to be addressed

7. PLOS authors have the option to publish the peer review history of their article (what does this mean?). If published, this will include your full peer review and any attached files.

Reviewer #2: No

---

## [Author Response · Author response to Decision Letter 1]

6 Aug 2022

Dear Dr. Louis-Felix Bersier, 

Thank you for allowing us the opportunity to revise our manuscript. Your comments were extremely helpful and made the manuscript much stronger. Based on their suggestions, we incorporated most suggestions to our manuscript. Below, you will find a letter detailing how we dealt with each comment you provided. 

We address each comment (C#) in individual answers (A#) to keep everything organized.

Editor Comments: 

C1: This level of precision is not necessary. I recommend rounding at unity here (21% to 69%)

A1: We have rounded these percentages. 

C2: Could be abridged as "tertiary-, secondary-, and primary burrowing species"

A2: This is a good suggestion, we have changed the text accordingly.

C3: delete

A3: We have deleted the words “the setup of” here. 

C4: I guess "... unable to report ..."

A4: Yes, you are correct. We have edited the text here. 

C5: Meaning not clear to me. If it indicates that this behavior occurs mostly during the night, it should not be part of the description in the table, but should be stated in the results.

A5: This statement (more exposed during the night) has been deleted because we agree that it is unclear. We meant to mean that when the crayfish exhibit the guard behavior during the night, they are typically more exposed out of their burrow portal. But we did not quantify this, so we chose to delete it from the text. 

C6: The images are very useful and raise a question since these two behaviors are very similar. As a non-specialist, I would like some words (and possibly a reference) that explain and justify this distinction.

A6: After some discussion, we agree with all of your comments regarding our denotation of these behaviors of “guard” and “rest” behaviors. We now refer to both of these behaviors as “rest” and separate them based on “rest-claws open/rest - open” and “rest-claws joined/rest – joined”. Although rest – closed may represent a guarding behavior, we believe that because this work is preliminary, we are keeping the language simple. The manuscript and figures have been changed accordingly. We believe that this language will properly describes the behaviors of the crayfishes would making any presumptions on the function of the posture. 

C1: There may be an unimodal relationships with these variable, reflecting optima. Did you check for this possibility ?

A1: We did check for this, but because there was not as much variation in the environment variables, we did not find any optimum relationship between activity and such environmental variables. 

C1: Personally, I would place the Tables 3 and 4 in the Supporting Information (only a suggestion).

A1: We have moved Table 3 and Table 4 to the SI as you have suggested. 

C1: This should be part of the Method section, e.g., on line 216, in the parenthesis "(

A1: We have moved this section the methods section as you suggested.

C1: Same remark as for the Abstract.

A1: All of our percentage values have been adjusted to be reported in the way that you suggest. 

C1: I do not see the correspondance between the percentages in the text and in Fig. 5. Please check if it is correct or provide better explanations (are the colors reversed in the figure ?)

A1: Thank for noticing this mistake. Yes, the colors on the figure legend should be reversed. in Figure 5. We have fixed this accordingly. Now the numbers and percentages add up correctly. 

C1: Again, I have problem with the values in the text and in Table 5. Crayfish 4 is the only one that foraged during the day. From Table 5, this happened only once, with a total of 360s, which is clearly not equal to 0.34 hr. Please explain.

A1: After checking back at our data, you are correct, we mistakenly calculated a larger percentage for the daytime activity of Crayfish 4. This change is now reflected in the manuscript and in the Figure 7, and Table 5. We re-checked the other crayfish as well and the results are correct for them. 

C1: Same remark as for Tables 3 and 4. The results are different but redundant with Fig. 7.

A1: We have moved this Table (Table 5) to the SI as you have suggested. 

C1: But Fig. 7 shows the opposite pattern! Table 5 support this.

A1: Thank you for catching this error. This was a typo and now correctly states that “The majority of their time was spent in the relax position during the night, and relaxing was more likely to occur during the night compared to during the day.”

C1: I think that "proportion" is more adequate here.

A1: We have reworded this to proportion based on your suggestion. 

C1: Again, I would place this Table in the SI, as the main information can be visualized in Fig. 8.

A1: We have moved this table to the SI based on your suggestions. 

C1: This relates to my question concerning Fig. 3 (line 183). Does reference 51 separates these two behaviors based on claws' position?

A1: See A6. 

C1: At this point, I am uncomfortable with your choice of words (relax and guard) for the two behaviors based on position of claws. Perhaps "guard - claws open" and "guard - claws joined" may be more appropriate?

A1: See A6.

---

## [Editor Report · Decision Letter 2]

11 Aug 2022

On the surface or down below: Field observations reveal a high degree of surface activity in a burrowing crayfish, the Little Brown Mudbug (Lacunicambarus thomai)

PONE-D-21-39709R2

Dear Dr. Diehl,

We’re pleased to inform you that your manuscript has been judged scientifically suitable for publication and will be formally accepted for publication once it meets all outstanding technical requirements.

Kind regards,

Louis-Felix Bersier, Ph.D.

Academic Editor

PLOS ONE

Additional Editor Comments (optional):

Thank you for your thorough consideration of my comments. I went through your corrected version and found that it is ready for publication. I just noted a probable mistake on lines 243-244: "(all r > 0.6)" rather than "(all r < 0.6)". Please check this before submitting your final document.
---

## [Editor Report · Acceptance letter]

2 Sep 2022

PONE-D-21-39709R2 

On the surface or down below: Field observations reveal a high degree of surface activity in a burrowing crayfish, the Little Brown Mudbug (*Lacunicambarus thomai*) 

Dear Dr. Diehl:

I'm pleased to inform you that your manuscript has been deemed suitable for publication in PLOS ONE. Congratulations! Your manuscript is now with our production department. 

Kind regards, 

on behalf of

Prof Louis-Felix Bersier 

Academic Editor

PLOS ONE